# Optimal Robust Classification Trees

**Nathan Justin,**[1] **Sina Aghaei,**[1] **Andrés Gómez,**[2] **Phebe Vayanos**[1]

[1]CAIS Center for Artificial Intelligence in Society
[2]Department of Industrial and Systems Engineering, Viterbi School of Engineering
[1,2]University of Southern California
Los Angeles, California 90007
{njustin,saghaei,gomezand,phebe.vayanos}@usc.edu

## Abstract

In many high-stakes domains, the data used to drive machine learning algorithms is noisy (due to e.g., the sensitive nature of the data being collected, limited resources available to validate the data, etc). This may cause a distribution shift to occur, where the distribution of the training data does not match the distribution of the testing data. In the presence of distribution shifts, any trained model can perform poorly in the testing phase. In this paper, motivated by the need for interpretability and robustness, we propose a mixed-integer optimization formulation and a tailored solution algorithm for learning optimal classification trees that are robust to adversarial perturbations in the data features. We evaluate the performance of our approach on numerous publicly available datasets, and compare the performance to a regularized, non-robust optimal tree. We show an increase of up to 14.16% in worst-case accuracy and increase of up to 4.72% in average-case accuracy across several data sets and distribution shifts from using our robust solution in comparison to the non-robust solution.

## 1 Introduction

Machine learning techniques are increasingly being used in high-stakes domains to assist humans in making important decisions. Within these applications, black box models that need explanation should be avoided as decisions made from these models may have a profound impact (Rudin 2019). That is, we need inherently interpretable models where decisions made can be simply understood and verified. One of the most interpretable models are classification trees, which are easily visualized and do not require extensive knowledge to use. Classification trees are a widely used model that takes the form of a binary tree. At each branching node, a test based off of the attributes of the given data sample is made, which dictates the next node visited. Then at an assignment node, a particular label is assigned to the data sample (Breiman et al. 2017).

However, as with many other machine learning models, classification trees are susceptible to distribution shifts. That is, the distribution of the training data and the testing data may be different, causing poor performance in deployment (Quiñonero-Candela et al. 2009). In high-stakes domains

where there is a need for interpretability, there must also be robustness against distribution shifts to ensure high-quality solutions under any realization of the training data.

### 1.1 Background and Related Work

Traditionally, classification trees are built using heuristic approaches since the problem of building optimal classification trees is $\mathcal{NP}$-hard (Breiman et al. 2017). But in settings where the quality of solutions is important, heuristic approaches may yield suboptimal solutions that are unacceptable for use in applications. Thus, to ensure a high-quality decision tree, mathematical optimization techniques, like mixed-integer optimization (MIO), have been developed for building optimal trees. Namely, Bertsimas and Dunn (2017) were the first to use MIO to build optimal decision trees. To combat the long run times for making optimal decision trees on large data sets, Verwer and Zhang (2019) create a binary linear programming formulation that has a run time independent of the amount of training samples. Aghaei, Gómez, and Vayanos (2021) build a strong, "flow-based" MIO formulation that greatly improves on solving times in comparison to other state-of-the-art optimal classification tree algorithms. MIO approaches for constructing decision trees have also allowed for several extensions. For example, Mišić (2020) formulates the problem of creating tree ensembles as a MIO problem, Aghaei, Azizi, and Vayanos (2019) create optimal and fair decision trees using MIO, and Jo et al. (2021) use MIO to build optimal prescriptive trees from observational data.

To account for the problem of distribution shifts, there exists both non-MIO and MIO approaches. One type of non-MIO method up-weights training samples that match the test set distribution and down-weights the training samples that differ from the test set (Shimodaira 2000; Bickel, Brückner, and Scheffer 2007). These methods usually define distribution shift as a biased sampling of training data, where assigning weights to training samples diminishes the effects of adversarial examples.

Another way to define a distribution shift is as an adversarial perturbation of the trained data that makes it differ from the test data. Motivated by this viewpoint, there have been several optimization-based methods to deal with distribution shifts. One of these methods is distributionally robust optimization, which combats the effects of distribution shifts

by performing well under an adversarial distribution of samples. Both Kuhn et al. (2019) and Sinha et al. (2020) in particular provide distributionally robust approaches to building machine learning models that perform well under an adversarial distribution of the training data, where the adversarial distribution is in some Wasserstein distance from the nominal distribution of the data.

Distributionally robust optimization requires an assumption on the distribution of the data available, which may not be a reasonable assumption to make. In the case where such an assumption cannot be reasonably made, robust optimization provides a framework to generate solutions that perform well in the worst-case perturbation, where the perturbation comes from a set of values without a probability distribution assumption imposed (Ben-Tal, El Ghaoui, and Nemirovski 2009). Many common machine learning models have been formulated as robust optimization problems to deal with uncertainty in data. For example, robust optimization has been used for creating robust support vector machines (Shivaswamy, Bhattacharyya, and Smola 2006; Bertsimas et al. 2019). Robust optimization has also been used to create artificial neural networks that are robust against adversarial perturbations of the data (Shaham, Yamada, and Negahban 2018).

In a similar spirit to these previous works, we propose using robust optimization to create a classification tree robust to distribution shifts. In an adversarial setting, we must decide the tree structure before observing the perturbation of the data. The perturbation of a sample, once unveiled, reveals whether the given tree structure correctly classifies the realization of the sample. That is, the classification of samples in a decision tree is dependent on both the tree structure and the realization of the uncertain parameter. Decision variables that indicate the correct classification of samples are called second-stage decisions, which can be modeled through two-stage robust optimization (Ben-Tal et al. 2004).

There have been several recent methods for robust classification trees. Namely, Vos and Verwer (2021) have a local search algorithm for decision trees robust against adversarial examples derived from a user-defined threat model. Bertsimas et al. (2019) have a robust optimization formulation for robust trees where the uncertainty set is modeled by restricting the norm of the perturbation parameter. Unlike our proposed method, Bertsimas et al. (2019) do not capture the dependent relationship between the perturbation of the covariates and the classification of training samples, as decisions made on the classification of training samples are made before the realization of the uncertain parameter. Thus, they cannot identify correctly classified data points in the realization of the worst-case perturbation. So, differing from previous work, we argue that the decision variables related to the classification of training samples should be modeled as second-stage decisions. And by modeling these variables as second-stage decisions, we obtain less conservative solutions than a single-stage approach to the same problem.

With these motivating factors in mind, we propose a two-stage, MIO method for learning optimal classification trees robust to distribution shifts in the data. Namely, we present a flow-based optimization problem, where we model uncertainty through a cost-and-budget framework. We then present a tailored Benders decomposition algorithm that solves this two-stage formulation to optimality. We evaluate the performance of our formulation on publicly available data sets for several problem instances to measure the effectiveness of our method in mitigating the adverse effects of distribution shifts.

## 2  Robust Tree Formulation

In this section, we present our formulation for a robust classification tree. We describe the structure of the classification tree, present the proposed two-stage formulation, and discuss our model of uncertainty.

### 2.1  Setup and Notation

Let $\{\mathbf{x}^i, y^i\}_{i \in \mathcal{I}}$ be the training set, where $\mathcal{I}$ is the index set for our training samples. The covariates are $\mathbf{x}^i \in \mathbb{Z}^{|\mathcal{F}|}$, where $\mathcal{F}$ is the set of features of our data, and $y^i$ is some label in a finite set $\mathcal{K}$. With a slight abuse of notation, we will let $\mathbf{x}$ denote the the vector concatenation of the rows of the $|\mathcal{I}| \times |\mathcal{F}|$ matrix of all training covariates, and $\boldsymbol{y}$ the $|\mathcal{I}|$-sized vector of all training labels. The training set $\{\mathbf{x}^i, y^i\}_{i \in \mathcal{I}}$ is used to determine the tree structure, which includes deciding what binary tests to perform and labels to predict. Thus, in the first stage of our problem, we decide the tree structure to maximize the number of correct classifications for the given training data.

Let $\boldsymbol{\xi}^i \in \mathbb{Z}^{|\mathcal{F}|}$ represent a perturbation of the covariate $\mathbf{x}^i$. We can only observe $\boldsymbol{\xi}^i \in \mathbb{Z}^{|\mathcal{F}|}$ after making our first stage decisions on the tree structure. So $\mathbf{x}^i + \boldsymbol{\xi}^i$ represent the realization of the training sample $i$ after determining the tree structure. We let the covariate and perturbation value at sample $i$ and feature $f$ be $x_f^i$ and $\xi_f^i$ respectively. Also, denote $\boldsymbol{\xi}$ as the vector concatenation of the rows of the $|\mathcal{I}| \times |\mathcal{F}|$ matrix of perturbations, and let $\Xi$ be our perturbation set that defines all possible $\boldsymbol{\xi}$.

As mentioned before, the second stage decisions are the classification of training samples, which occurs after deciding the tree structure and observing the worst-case perturbation. To classify sample $i$ after perturbation, we must perform the series of binary tests for covariate $\mathbf{x}^i + \boldsymbol{\xi}^i$ from the tree decided on in the first stage. The binary test uses a threshold $\theta$ such that if $x_f^i + \xi_f^i \leq \theta$, then sample $i$ travels to the left child. Likewise, if $x_f^i + \xi_f^i \geq \theta + 1$, then sample $i$ travels to the right child. Letting $c_f$ and $d_f$ be the lower and upper bound of realized values for feature $f$ respectively, we define
$$\Theta(f) := \{\theta \in \mathbb{Z} \mid c_f \leq \theta < d_f\}$$
as the set of possible binary test threshold values for feature $f$.

### 2.2  The Two-Stage Problem

We will set up our robust formulation based on the non-robust classification tree outlined by (Aghaei, Gómez, and Vayanos 2021), where a binary classification tree is represented by a directed graph. The model starts with a depth $d$ binary tree, where the internal nodes are in the set $\mathcal{N}$ and

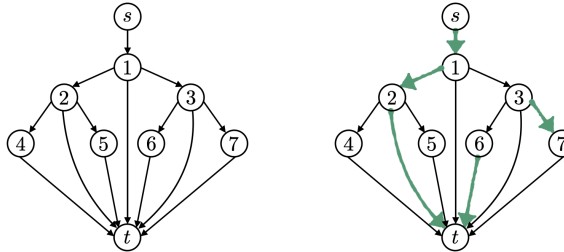

Figure 1: The left graph shows a classification tree with depth 2. Nodes $s$ and $t$ are the source and sink nodes respectively, $\mathcal{N} = \{1, 2, 3\}$ and $\mathcal{L} = \{4, 5, 6, 7\}$. The right graph shows an example of induced graph of a particular sample, where the green, bold edges are the subset of edges included in the induced graph. For this particular induced graph, nodes 1 and 3 are branching nodes, where the sample would be routed left and right, respectively. Nodes 2 and 6 assign the correct label, and node 7 does not assign the correct label. The maximum flow from $s$ to $t$ is 1 in this induced graph, indicating a correct classification.

the leaf nodes are in the set $\mathcal{L}$. A node in $\mathcal{N}$ can either be a branching node where a binary test is performed, or an assignment node where a classification of a sample is made. Nodes in $\mathcal{L}$ can only be assignment nodes. There are $2^d - 1$ nodes in $\mathcal{N}$ and $2^d$ nodes in $\mathcal{L}$, and we number each node from 1 to $2^{d+1} - 1$ in a breadth-first search pattern. We then augment this tree by adding a single source node $s$ connected to the root node of the tree, and a sink node $t$ connected to each node in $\mathcal{N} \cup \mathcal{L}$. By adding a source and sink node, we say that any data sample $i$ travels from the source $s$ through the tree based on $\mathbf{x}^i$ and reaches the sink if and only if the datapoint is correctly classified. Lastly, we denote the left and right child of node $n \in \mathcal{N}$ as $l(n)$ and $r(n)$ respectively, and also denote the ancestor of any node $n \in \mathcal{N} \cup \mathcal{L}$ as $a(n)$.

To determine whether a data sample $i$ is correctly classified by a given tree, we consider an induced graph of the original directed graph for data sample $i$. In this induced graph, we keep every node in $\mathcal{N} \cup \mathcal{L}$, the source $s$, and the sink $t$. For every branching node, we remove the edge leading to the sink node $t$ and the edge that fails the binary test based on the value of $\mathbf{x}^i + \boldsymbol{\xi}^i$. And for every assignment node, we include the edge leading to $t$ only if the assigned class of that node is $y^i$, and exclude all other edges leaving the assignment node. Lastly, we include the edge from the source $s$ to the root node 1. Therefore, a maximum flow of 1 from $s$ to $t$ of this induced graph means that data sample $i$ would be correctly classified. Figure 1 illustrates an induced graph.

With the flow graph setup, we propose a two-stage formulation for creating robust classification trees. In the first stage, we decide the structure of the tree. Let $b_{nf\theta}$, defined over all $n \in \mathcal{N}$, $f \in \mathcal{F}$, and $\theta \in \Theta(f)$ be a binary variable that denotes the branching decisions. If $b_{nf\theta} = 1$, then node $n$ is a branching node, where the binary test is on feature $f$ with threshold $\theta$. We also let $w_{nk}$, defined over all

$n \in \mathcal{N} \cup \mathcal{L}$ and all $k \in \mathcal{K}$, be a binary variable that denotes the assignment decisions. If $w_{nk} = 1$, then node $n$ is an assignment node with assignment label $k$. We will denote $\mathbf{b}$ and $\mathbf{w}$ as the collection of $b_{nf\theta}$ and $w_{nk}$ variables respectively.

Given a value of $\mathbf{b}$ and $\mathbf{w}$, we find the perturbation in $\Xi$ that results in the minimum number of correctly classified points, which we will call the worst-case perturbation. In the second stage, after observing the worst-case perturbation of our covariates $\boldsymbol{\xi}$ from the set $\Xi$ given a certain tree structure, we classify each of our points based on our tree. Let $z^i_{n,m}$ indicate whether data point $i$ flows down the edge between $n$ and $m$ and is correctly classified by the tree for $n \in \mathcal{N} \cup \mathcal{L} \cup \{s\}$ and $m \in \mathcal{N} \cup \mathcal{L} \cup \{t\}$ under the worst-case perturbation. We will let $\mathbf{z}$ be the vector concatenation of the rows of the $|\mathcal{I}| \times (2^{d+2} - 2)$ matrix of $z^i_{n,m}$ values with rows corresponding to data sample $i$ and columns representing edge $(n, m)$ for $n = a(m)$. Note that $z^i_{n,m}$ are the decision variables of a maximum flow problem, where in the induced graph from data sample $i$, $z^i_{n,m}$ is 1 if and only if the maximum flow is 1 and the flow goes from node $n$ to node $m$. Therefore, if there exists an $n \in \mathcal{N} \cup \mathcal{L}$ such that $z^i_{n,t} = 1$, then sample $i$ is correctly classified by our tree after observing the worst-case perturbation.

Our two-stage approach to defining the variables $\mathbf{b}$, $\mathbf{w}$, $\boldsymbol{\xi}$, and $\mathbf{z}$ leads us to the following formulation for a robust classification tree:

$$\max_{\mathbf{b}, \mathbf{w}} \min_{\boldsymbol{\xi} \in \Xi} \max_{\mathbf{z} \in \mathcal{Z}(\mathbf{b}, \mathbf{w}, \boldsymbol{\xi})} \sum_{i \in \mathcal{I}} \sum_{n \in \mathcal{N} \cup \mathcal{L}} z^i_{n,t} \qquad (1a)$$

$$\text{s.t.} \sum_{f \in \mathcal{F}} \sum_{\theta \in \Theta(f)} b_{nf\theta} + \sum_{k \in \mathcal{K}} w_{nk} = 1 \quad \forall n \in \mathcal{N} \quad (1b)$$

$$\sum_{k \in \mathcal{K}} w_{nk} = 1 \qquad \forall n \in \mathcal{L} \quad (1c)$$

$$b_{nf\theta} \in \{0, 1\} \quad \forall n \in \mathcal{N}, f \in \mathcal{F}, \theta \in \Theta(f) \quad (1d)$$

$$w_{nk} \in \{0, 1\} \qquad \forall n \in \mathcal{N} \cup \mathcal{L}, k \in \mathcal{K}, \quad (1e)$$

where the set $\mathcal{Z}$ is defined as

$$\mathcal{Z}(\mathbf{b}, \mathbf{w}, \boldsymbol{\xi}) := \{\mathbf{z} \in \{0, 1\}^{|\mathcal{I}| \times (2^{d+2} - 2)} :$$

$$z^i_{n,l(n)} \leq \sum_{f \in \mathcal{F}} \sum_{\substack{\theta \in \Theta(f): \\ x^i_f + \xi^i_f \leq \theta}} b_{nf\theta} \qquad \forall i \in \mathcal{I}, n \in \mathcal{N}, \quad (2a)$$

$$z^i_{n,r(n)} \leq \sum_{f \in \mathcal{F}} \sum_{\substack{\theta \in \Theta(f): \\ x^i_f + \xi^i_f \geq \theta + 1}} b_{nf\theta} \qquad \forall i \in \mathcal{I}, n \in \mathcal{N}, \quad (2b)$$

$$z^i_{a(n),n} = z^i_{n,l(n)} + z^i_{n,r(n)} + z^i_{n,t} \quad \forall i \in \mathcal{I}, n \in \mathcal{N}, \quad (2c)$$

$$z^i_{a(n),n} = z^i_{n,t} \qquad \forall i \in \mathcal{I}, n \in \mathcal{L}, \quad (2d)$$

$$z^i_{n,t} \leq w_{n,y^i} \qquad \forall i \in \mathcal{I}, n \in \mathcal{N} \cup \mathcal{L} \quad (2e)$$

$$\}.$$

The objective function in (1) maximizes the number of correctly classified training samples in the worst-case perturbation. The constraint (1b) states that each internal node must either classify a point or must be a binary test with some

threshold $\theta$. The constraint (1c) states that each leaf node must classify a point.

The set (2) describes the maximum flow constraints for each sample's induced graph. Constraints (2a) and (2b) are capacity constraints that control the flow of samples in the induced graph based on $\mathbf{x} + \boldsymbol{\xi}$ and the tree structure. Constraints (2c) and (2d) are flow conservation constraints. Lastly, constraint (2e) blocks any flow to the sink if the node is either not an assignment node or the assignment is incorrect.

## 2.3 The Uncertainty Set

We consider uncertainty sets defined as follows. Let $\gamma_f^i \in \mathbb{R}$ be the cost of perturbing $x_f^i$ by one. Thus, $\gamma_f^i |\xi_f^i|$ is the total cost of perturbing $x_f^i$ to $x_f^i + \xi_f^i$. Letting $\epsilon$ be the total allowable budget of uncertainty across data samples, we define the following uncertainty set:

$$\Xi := \left\{ \boldsymbol{\xi} \in \mathbb{Z}^{|\mathcal{I}| \times |\mathcal{F}|} : \sum_{i \in \mathcal{I}} \sum_{f \in \mathcal{F}} \gamma_f^i |\xi_f^i| \leq \epsilon \right\}. \quad (3)$$

As we will show later, a tailored solution method of Formulation (1) can be made if uncertainty is defined by set (3), and there exists a connection between (3) and hypothesis testing.

# 3 Solution Method

We now present a method of solving problem (1) through a reformulation that can leverage existing, off-the-shelf mixed-integer linear programming solvers.

## 3.1 Reformulating the Two-Stage Problem

We solve our two-stage optimization problem by first taking the dual of the inner maximization problem. Recall that the inner maximization problem is a maximum flow problem; therefore, the dual of the inner maximization problem will yield an inner minimum cut problem (Vazirani 2001). Note that strong duality holds, and therefore taking the dual of the inner problem of (1) will yield a reformulation with equal optimal objective values, and thus optimal tree structure variables $\mathbf{b}$ and $\mathbf{w}$ for both problems.

Let $q_{n,m}^i$ be the binary dual variable that equals 1 if and only if in the induced graph for data sample $i$ after perturbation, the edge that connects nodes $n \in \mathcal{N} \cup \{s\}$ and $m \in \mathcal{N} \cup \mathcal{L} \cup \{t\}$ is in the minimum cut. We write $\mathbf{q}$ as the vector concatenation of the rows of the $|\mathcal{I}| \times (2^{d+2} - 2)$ matrix of $q_{n,m}^i$ values with rows corresponding to data sample $i$ and columns representing edge $(n, m)$. We also define $p_n^i$ to be a binary variable that equals 1 if and only if in the induced graph of data sample $i \in \mathcal{I}$, the node $n \in \mathcal{N} \cup \mathcal{L} \cup \{s\}$ is in the source set. Letting $\mathcal{Q}$ be the set of all possible values of

$\mathbf{q}$, we then have

$$\mathcal{Q} := \{ \mathbf{q} \in \{0, 1\}^{|\mathcal{I}| \times (2^{d+2} - 2)} :$$
$$\exists p_n^i \in \{0, 1\} \qquad \forall i \in \mathcal{I}, n \in \mathcal{N} \cup \mathcal{L} \cup \{s\}, \quad (4a)$$
$$q_{n,l(n)}^i - p_n^i + p_{l(n)}^i \geq 0 \qquad \forall i \in \mathcal{I}, n \in \mathcal{N}, \quad (4b)$$
$$q_{n,r(n)}^i - p_n^i + p_{r(n)}^i \geq 0 \qquad \forall i \in \mathcal{I}, n \in \mathcal{N}, \quad (4c)$$
$$q_{s,1}^i + p_1^i \geq 1 \qquad \forall i \in \mathcal{I}, \quad (4d)$$
$$- p_n^i + q_{n,t}^i \geq 0 \qquad \forall i \in \mathcal{I}, n \in \mathcal{N} \cup \mathcal{L}, \quad (4e)$$
$$\}.$$

Constraints (4b) ensures that if the node $n \in \mathcal{N}$ is in the source set, then either its left child is also in the source set or the edge between $n$ and its left child are in the cut. Constraint (4c) is analogous to (4b), but with the right child of node $n \in \mathcal{N}$. Constraint (4d) states that either the root node is in the source set or the edge from the source to the root node is in the cut. Lastly, constraint (4e) ensures that for any $n \in \mathcal{N} \cup \mathcal{L}$, if $n$ is in the source set, then the edge from $n$ to the sink must be in the cut.

Then, taking the dual of the inner maximization problem in (1) gives the following single-stage formulation:

$$\max_{\mathbf{b}, \mathbf{w}} \min_{\mathbf{q} \in \mathcal{Q}, \boldsymbol{\xi} \in \Xi} \sum_{i \in \mathcal{I}} \sum_{n \in \mathcal{N}} \sum_{f \in \mathcal{F}} \sum_{\substack{\theta \in \Theta(f): \\ x_f^i + \xi_f^i \leq \theta}} q_{n,l(n)}^i b_{nf\theta}$$

$$+ \sum_{i \in \mathcal{I}} \sum_{n \in \mathcal{N}} \sum_{f \in \mathcal{F}} \sum_{\substack{\theta \in \Theta(f): \\ x_f^i + \xi_f^i \geq \theta+1}} q_{n,r(n)}^i b_{nf\theta}$$

$$+ \sum_{i \in \mathcal{I}} \sum_{n \in \mathcal{N} \cup \mathcal{L}} q_{n,t}^i w_{n,y^i} + \sum_{i \in \mathcal{I}} q_{s,1}^i \quad (5a)$$

$$\text{s.t.} \sum_{f \in \mathcal{F}} \sum_{\theta \in \Theta(f)} b_{nf\theta} + \sum_{k \in \mathcal{K}} w_{nk} = 1 \quad \forall n \in \mathcal{N} \quad (5b)$$

$$\sum_{k \in \mathcal{K}} w_{nk} = 1 \qquad \forall n \in \mathcal{L} \quad (5c)$$

$$b_{nf\theta} \in \{0, 1\} \quad \forall n \in \mathcal{N}, f \in \mathcal{F}, \theta \in \Theta(f) \quad (5d)$$

$$w_{nk} \in \{0, 1\} \qquad \forall n \in \mathcal{N} \cup \mathcal{L}, k \in \mathcal{K} \quad (5e)$$

where $\mathcal{Q}$ is defined by (4) and constraints (5b) and (5c) are the same as constraints (1b) and (1c) respectively. As mentioned before, strong duality holds between Formulations (1) and (5) since strong duality holds between the maximum flow and minimum cut problems.

## 3.2 Solving the Single-Stage Reformulation

We can obtain a mixed-integer linear program equivalent to (5) by doing a hypograph reformulation. However, a hypograph reformulation would introduce an extremely large of constraints. A common approach to solving the reformulation is to use a tailored Benders decomposition algorithm, which we describe here.

The master problem decides the tree structure given its current constraints. We thus have the following initial master

problem:

$$\max_{\mathbf{b},\mathbf{w},\mathbf{t}} \sum_{i \in \mathcal{I}} t_i \tag{6a}$$

$$\text{s.t.} \sum_{f \in \mathcal{F}} \sum_{\theta \in \Theta(f)} b_{nf\theta} + \sum_{k \in \mathcal{K}} w_{nk} = 1 \quad \forall n \in \mathcal{N} \tag{6b}$$

$$\sum_{k \in \mathcal{K}} w_{nk} = 1 \qquad \forall n \in \mathcal{L} \tag{6c}$$

$$t_i \leq 1 \qquad \forall i \in \mathcal{I} \tag{6d}$$

$$b_{nf\theta} \in \{0,1\} \quad \forall n \in \mathcal{N}, f \in \mathcal{F}, \theta \in \Theta(f) \tag{6e}$$

$$w_{nk} \in \{0,1\} \qquad \forall n \in \mathcal{N} \cup \mathcal{L}, k \in \mathcal{K} \tag{6f}$$

where $t_i$ comes from the hypograph of the inner sum of objective function (5a) for a particular $i \in \mathcal{I}$. We add the constraint (6d) to ensure that the initial problem is bounded.

The goal for the subproblem is, given certain values of $\mathbf{b}$, $\mathbf{w}$, and $\mathbf{t}$ that describe a specific tree structure, find a perturbation in $\Xi$ that reduces the number of correctly classified samples the most. After finding the minimum cut of the induced graph for each data sample after deciding the perturbation, we add a constraint of the form

$$\sum_{i \in \mathcal{I}} t_i \leq \sum_{i \in \mathcal{I}} \sum_{n \in \mathcal{N}} \sum_{f \in \mathcal{F}} \sum_{\substack{\theta \in \Theta(f): \\ x_f^i + \xi_f^i \leq \theta}} q_{n,l(n)}^i b_{nf\theta}$$

$$+ \sum_{i \in \mathcal{I}} \sum_{n \in \mathcal{N}} \sum_{f \in \mathcal{F}} \sum_{\substack{\theta \in \Theta(f): \\ x_f^i + \xi_f^i \geq \theta+1}} q_{n,r(n)}^i b_{nf\theta} \tag{7}$$

$$+ \sum_{i \in \mathcal{I}} \sum_{n \in \mathcal{N} \cup \mathcal{L}} q_{n,t}^i w_{n,y^i} + \sum_{i \in \mathcal{I}} q_{s,1}^i$$

where we substitute the variables $\boldsymbol{\xi}$ and $\mathbf{q}$ with the perturbation and minimum cuts.

### 3.3 The Subproblem

We will now describe the procedure for the subproblem to find the perturbation and minimum cuts that will yield a violated constraint of the form (7) if a violated constraint exists.

For each data sample that is correctly classified by the tree given by the master problem (6), we first find the lowest-cost perturbation $\boldsymbol{\xi}^i$ for the single sample that would cause it to be misclassified. To do this, we set up a shortest path problem. The weighted graph of the shortest path problem is created from the flow-based tree returned by the master problem, and is constructed with the following procedure:

1. The edge from $s$ to 1 (the root of the decision tree) has path cost 0.
2. For each $n \in \mathcal{N} \cup \mathcal{L}$, if there exists a $k \in \mathcal{K}$ such that $w_{nk} = 1$, and $k \neq y^i$, then we have a 0 path cost from $n$ to $t$. All other edges coming into $t$ have infinite path cost.
3. For each $n \in \mathcal{N}$, if there exists an $f \in \mathcal{F}$ such that $b_{nf} = 1$, then...
   (a) if $x_f^i = 0$, add an edge from $n$ to $l(n)$ with 0 weight, and add an edge from $n$ to $r(n)$ with $\gamma_f$ weight.
   (b) if $x_f^i = 1$, add an edge from $n$ to $r(n)$ with 0 weight, and add an edge from $n$ to $l(n)$ with $\gamma_f$ weight.

By finding the shortest path from $s$ to $t$ for the weighted graph derived from data sample $i$, we find the path with the smallest total cost of perturbation that would misclassify the point $i$. That is, we use the shortest path to see what perturbation $\boldsymbol{\xi}^i$ would misclassify $\mathbf{x}^i$ with the smallest cost.

Once we find the lowest-cost perturbation that would misclassify every sample, we choose the largest subset of these training samples whose total cost of perturbation to misclassify each sample is less than the allowed budget of uncertainty. Through this procedure, we find the value of $\boldsymbol{\xi}$ that misclassifies the most number of points given the current tree.

Note that the right hand side of the constraint (7) gives the count of the the number of correctly classified points for a certain $\mathbf{b}$, $\mathbf{w}$, $\boldsymbol{\xi}$, and $\mathbf{q}$. Therefore, for dataset $\mathbf{x} + \boldsymbol{\xi}$, if the number of correctly classified points is less than the optimal value of $\sum_{i \in I} t_i$ from the master problem, then we know that there exists a constraint of the form (7) that is violated. Otherwise, there are no violated constraints of the form (7), which indicates the optimality of the current solution.

In the case of finding a violated constraint, we now would like to obtain the values of $\mathbf{q}$, the variables associated with the minimum cut problem. To do this, we need to find for each sample $i \in \mathcal{I}$ the set of edges in a minimum cut given the path of the data point $\mathbf{x}^i + \boldsymbol{\xi}^i$, where the value of the cut is 1 if $\mathbf{x}^i + \boldsymbol{\xi}^i$ is correctly classified and 0 otherwise. The simplest way to construct this minimum cut is for each $i \in \mathcal{I}$, we follow the path of $\mathbf{x}^i + \boldsymbol{\xi}^i$. At each node visited in this path, we first include to the minimum cut any edges outgoing from that node that are not traversed. And at the assignment node, we also include the edge going from the assignment node to the sink $t$. By doing this procedure for all training samples, we obtain the value of $\mathbf{q}$ describing all minimum cuts.

By finding the value of $\boldsymbol{\xi}$ and an associated $\mathbf{q}$, we can use these values in (7) to obtain the most restrictive violated constraint for a given tree, which we add back to the master problem. We summarize our approach in Algorithm 1.

## 4 Statistical Connections

Here, we explore how the uncertainty set described in (3) connects to hypothesis testing. Let $q_f^\zeta \in (0,1]$ be the probability that the realization of the data at feature $f$ as decided by our uncertainty set perturbs the nominal data at feature $f$ by $\zeta \in \mathbb{Z}$. We will impose the assumption that the perturbations follow a geometric distribution. More specifically,

$$q_f^\zeta = (0.5)^{\mathbb{I}[\zeta \neq 0]} q_f (1 - q_f)^{|\zeta|} \tag{8}$$

for some $q_f \in (0,1]$ (where the $(0.5)^{\mathbb{I}[\zeta \neq 0]}$ multiplier imposes a symmetry between positive and negative values of the perturbation).

We will set up a likelihood ratio test with threshold $\lambda^{|\mathcal{I}|}$ for $\lambda \in [0,1]$, where we add the exponential of $|\mathcal{I}|$ for ease of comparison across different data sets with different number of training samples. Our null hypothesis will be that a given perturbation of our data comes from the distribution of perturbations given by the chosen $q_f^\zeta \in (0,1]$. Then, we

Algorithm 1: Solution Method to formulation (5)

---

**Input**: training set indexed by $\mathcal{I}$ with features $\mathcal{F}$ and labels $\mathcal{K}$, range of test thresholds $\Theta(f)$, tree depth $d$, uncertainty set parameters $\gamma$ and $\epsilon$

**Output**: The optimal robust tree represented by $\mathcal{T}^* = (\mathbf{b}^*, \mathbf{w}^*)$

1: **while** no tree $\mathcal{T}$ returned **do**
2:     Solve the master problem (6) with any added constraints, obtain tree $\mathcal{T} = (\mathbf{b}^*, \mathbf{w}^*)$, and $\mathbf{t}^*$
3:     Find the lowest cost $\boldsymbol{\xi}$ that causes the most number of samples to be misclassified in $\mathcal{T}$ to obtain $\boldsymbol{\xi}^*$
4:     **if** $\sum_{i \in \mathcal{I}} t_i \leq$ number of correctly classified samples of $\mathbf{x} + \boldsymbol{\xi}^*$ given $\mathcal{T}$ **then**
5:         **return** $\mathcal{T}$
6:     **else**
7:         Find $\mathbf{q}^*$ by finding a minimum cut for each $i \in \mathcal{I}$ based on $\mathbf{x} + \boldsymbol{\xi}^*$ and $\mathcal{T}$
8:         Use values of $\boldsymbol{\xi}^*$ and $\mathbf{q}^*$ to create constraint (7) to add to the master problem.
9:     **end if**
10: **end while**

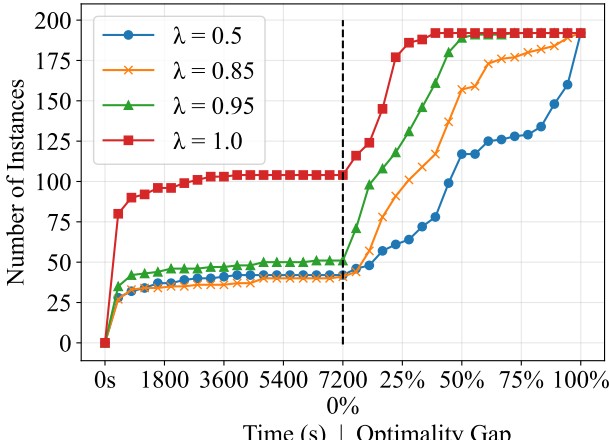

Figure 2: This graph shows the number of instances solved across times and optimality gaps when the time limit of 7200 seconds is reached for several values of $\lambda$. The case of $\lambda = 1.0$ is the regularized tree with an empty uncertainty set.

set up a likelihood ratio test where we fail to reject the null hypothesis if

$$\frac{\prod_{i \in \mathcal{I}} \prod_{f \in \mathcal{F}} \prod_{\zeta=-\infty}^{\infty} \left(q_f^\zeta\right)^{\mathbb{I}[\xi_f^i == \zeta]}}{\prod_{i \in \mathcal{I}} \left[\prod_{f \in \mathcal{F}} q_f^0\right]} \geq \lambda^{|\mathcal{I}|}, \quad (9)$$

where the numerator of the left hand side is the maximum likelihood of a given perturbation $\boldsymbol{\xi}$, and the denominator of the left hand side is the likelihood under the null hypothesis. Using the assumption that $q_f^\zeta$ follows (8), we can reduce the hypothesis test in (9) into

$$\sum_{i \in \mathcal{I}} \sum_{f \in \mathcal{F}} |\xi_f^i| \log\left(\frac{1}{1 - q_f}\right) \leq -|\mathcal{I}| \log \lambda. \quad (10)$$

We say that if a particular $\boldsymbol{\xi}$ lies within the region where we fail to reject the null hypothesis, then it is part of our perturbation set. That is, using the notation from the perturbation set defined in (3), letting $\gamma_f^i = \log\left(\frac{1}{1-q_f}\right)$ and $\epsilon = -|\mathcal{I}| \log \lambda$ yields an uncertainty set with a direct relationship to the probabilities of certainty for each feature.

## 5 Experiments

We evaluate our approach on 12 datasets from the UCI Machine Learning Repository (Dua and Graff 2017). We used datasets that can be encoded into either binary or integer-valued features. The number of samples ranged from 124 to 3196 and the number of features from 4 to 36. For each data set, we construct a robust classification tree from our method using a synthetic uncertainty set where for different problem instances, we choose different levels of uncertainty in the features and budgets of uncertainty. We utilize the hypothesis testing framework as described by (10), where we define

$q_f$ by sampling the probability of certainty from a normal distribution with a particular mean we set and a standard deviation of 0.2. The means of this normal distribution included 0.6, 0.7, 0.8, and 0.9. We also chose different values of our budget by setting $\lambda$ to be 0.5, 0.75, 0.85, 0.9, 0.95, 0.97, and 0.99. For every data set and uncertainty set, we tested with tree depths of 2, 3, 4 and 5.

For each instance, we randomly split the data set into 80% training data, 20% testing data. We then ran our algorithm to obtain a robust classification tree with a time limit of 7200 seconds. For comparison, we used our model to create a non-robust tree as well by setting the budget of uncertainty to 0 (i.e. $\lambda = 1$), and tuned a regularization parameter for the non-robust tree. The regularization term penalized the objective for every branching node to yield the following objective in the master problem of our algorithm:

$$\max_{\mathbf{b}, \mathbf{w}, \mathbf{t}} (1 - R) \sum_{i \in \mathcal{I}} t_i - R \sum_{i \in \mathcal{I}} \sum_{n \in \mathcal{N}} \sum_{\Theta(f)} b_{nf\theta}$$

where $R \in [0, 1]$ is the tuned regularization parameter. Note that we do not add a regularization parameter to our robust model, as robust optimization has an equivalence with regularization and so adding a regularization term is redundant (Bertsimas and Copenhaver 2018). We summarize the computation time across all instances in Figure 2. As we expected, the larger the uncertainty set, the longer it takes for the formulation to solve to optimality.

To test our model's robustness against distribution shifts, we perturbed the test data in 5000 different ways, where for each perturbation we found the test accuracy from our robust tree. We first perturbed the data based on the expected distribution of perturbations. That is, for the collection of $q_f$ values for every $f \in \mathcal{F}$ used to construct an uncertainty set based off of (10), we perturb the data based on the distribution described in (8).

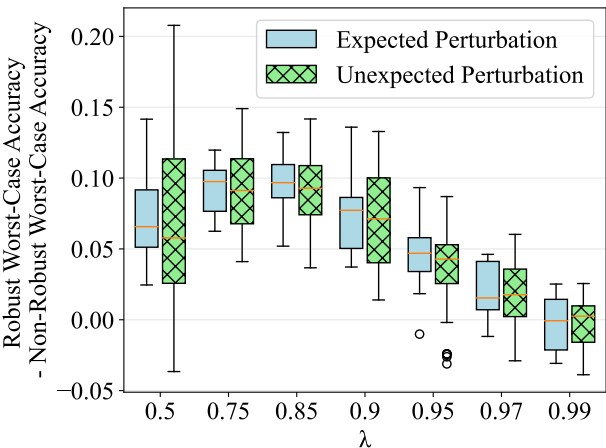

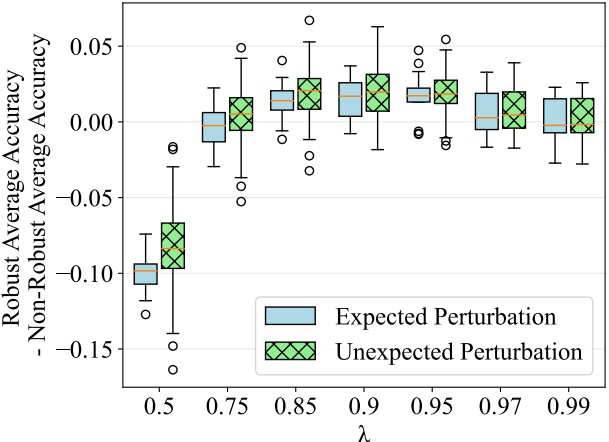

Figure 3: These boxplots show the distribution across problem instances of the gain in worst-case accuracy from using a robust tree versus a non-robust, regularized tree across different values of $\lambda$. We also show the distribution of the gain in worst-case accuracy in the case where perturbations of our data are not as we expect.

Figure 4: These boxplots show the distribution across problem instances of the gain in average test accuracy from using a robust tree versus a non-robust, regularized tree across different values of $\lambda$. We also show this gain in average accuracy in the case where perturbations of our data are not what we expect.

In order to measure the robustness of our model based on unexpected perturbations of the data, we also repeat the same process but for values of $q_f$ different than what we gave our model. First, we shifted each $q_f$ value down 0.2, then perturb our test data in 5000 different ways based on these new values of $q_f$. We do the same procedure but with $q_f$ shifted down by 0.1 and up by 0.1. In a similar fashion, we also uniformly sampled a new $q_f$ value for each feature in a neighborhood of radius 0.05 of the original expected $q_f$ value, and perturbed the test data in 5000 different ways with the new $q_f$ values. We do the same procedure for the radii of the neighborhoods 0.1, 0.15, and 0.2,

For each set of perturbations of the test data, we measure the worst-case accuracy by finding the lowest accuracy from all perturbations we made for a single set of $q_f$ values, and measure the average accuracy by averaging over the accuracy over all perturbations for a single set of $q_f$ values. We compile the gain in worst-case and average-case performance from using our robust tree versus using a regularized, non-robust tree for every problem instance and perturbation of our data, giving us a distribution of worst-case and average case gains in performance that are summarized in Figures 3 and 4, respectively.

From the figures, we see that our robust tree model in general has both higher worst-case and average-case accuracy than a non-robust model when there exists distribution shifts in the data. We also see that there is a range of values of $\lambda$ that seem to perform well over other values (namely 0.85). This shows us that if the budget of uncertainty is too small, then we do not allow enough room to hedge against distribution shifts in our uncertainty set. But if the budget of uncertainty is too large, then we become over-conservative and perform poorly for any perturbation of our test data. We also see that there is little difference between the gains in accu-

racy in instances where the perturbation of our data is as we expected versus when the perturbation is not as we expect. This indicates that even if we misspecify our model, we still obtain a classification tree robust to any kind of distribution shift within a reasonable range of our expected distribution shift. Overall, we see that an important factor in determining the performance of our model is the budget of uncertainty, which can be easily tuned to create an effective robust tree.

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

## 6   Acknowledgments

P. Vayanos and S. Aghaei gratefully acknowledge support from the Hilton C. Foundation, the Homeless Policy Research Institute, the Home for Good foundation under the "C.E.S. Triage Tool Research & Refinement" grant. P. Vayanos and N. Justin are grateful for the support of the National Science Foundation, under CAREER grant 2046230. A. Gómez is funded in part by the National Science Foundation under grant 1930582.
