# OpenReview forum: "Optimal Robust Classification Trees"
_AAAI.org/2022/Workshop/AdvML — AAAI-22 AdvML Workshop LongPaper_

### Official Review · Reviewer_d2sp · 2021-11-25
**Interesting Approach but needs better evaluation**

**Rating:** 6
**Confidence:** 3

**Review:**

**Summary**:

The paper introduces a method to build optimal decision trees that are robust to adversarial perturbations. The authors present a cost and budget framework which can be used to obtain an optimal solution using a modified form of Benders decomposition. They conduct evaluation on several public datasets and demonstrate the robustness of their model against shifts in distribution.

**PROS**

1. Decision trees are widely used in the industry because they are interpretable models; this work will be useful in defending against adversarial attacks in critical operations like risk, fraud and security.
2. The authors have provided a strong mathematical foundation to their approach. In particular, I am impressed with the proofs they have presented, which are thorough and detailed.
3. The authors have performed significant testing and benchmarking to evaluate their approach.

**CONS**
1. Although the testing is non-trivial, the authors have not described which datasets they have used for testing their approach. The UCI repository has a number of datasets, with a wide variety of features (categorical, continuous, non-normalized, standardized, etc). Since the results will depend on the nature of the features, it is important to know what datasets were used.
2. The authors have considered 7200 seconds as a time limit for constructing the optimal tree. The paper should have an evaluation that describes how the performance is affected as more time is permitted for training. This is because in many industry settings, model training is conducted asynchronously and offline, and hence more time may be provided.
3. As is common with research on adversarial attacks and robustness, the paper should ideally provide an example of perturbed data points (preferably on a dataset with a small number of explainable features) that are misclassified by a standard (non-robust) decision tree but correctly classified using your approach.

---

### Official Review · Reviewer_iYNi · 2021-11-29
**A Two-Stage Problem Formulation for Deriving Robust Classification Tree**

**Rating:** 6
**Confidence:** 2

**Review:**

This paper proposes a novel framework to formalize generating robust classification tree as a two-stage optimization problem. The authors address the generation problem as optimizing the decision variables after calculating the worst perturbation sets.

Despite the performance improvement of proposed model, the method seems to be inefficient when scaling to large datasets, because it can be very expensive to find worst perturbation sets within such large datasets. Also, the baseline addressed in this paper is merely non-robust one, it would be more comprehensive to include other robust tree generation methods for comparison.

---

### Decision · Program_Chairs · 2021-12-01

**Decision:**

Accept (Long Paper)

**Comment:**

Both reviewers give positive ratings on this paper. Thus it is accepted as a long paper. Please address the reviewers' comments in camera-ready.